# Performance of Winter Wheat (*Triticum aestivum*) Depending on Fungicide Application and Nitrogen Top-Dressing Rate

**Zinta Gaile [1,\*], Biruta Bankina [1], Ieva Pluduma-Paunina [1], Linda Sterna [1], Gunita Bimsteine [1], Agrita Svarta [1], Janis Kaneps [1], Irina Arhipova [2] and Aigars Sutka [3]**

1   Faculty of Agriculture, Latvia University of Life Sciences and Technologies, 2 Liela Str., LV-3001 Jelgava, Latvia
2   Faculty of Information Technologies, Latvia University of Life Sciences and Technologies, 2 Liela Str., LV-3001 Jelgava, Latvia
3   AKPC, Ltd., 109 Vienibas Gatve, LV-1058 Riga, Latvia
\*   Correspondence: zinta.gaile@lbtu.lv

**Abstract:** Winter wheat (*Triticum aestivum*) is a crop of which production is associated with rather large investments for nitrogen fertilization and disease control. The aim of this study was to estimate the effect of five variants of fungicide application and four levels of N (nitrogen) top-dressing rate on the yield and grain quality of winter wheat. Field trials were carried out in Latvia (56° 31′ N; 23° 42′ E) for four seasons. Grain yield and quality depended significantly on the conditions of the trial year, as three of them were characterized by drought in varying degrees. Although the average four-year grain yield increased significantly in all fungicide application variants, the effect of this factor was different in individual years. The application of fungicides increased the yield significantly in one year, decreased significantly in another year, while it had no significant effect on the yield in remaining two seasons. The enhancement of N top-dressing rate increased the grain yield significantly every year. The interaction between both examined factors was significant; however, the use of higher N rates not always means that also spraying with fungicides has to be more intensive. A clear effect of fungicide application was observed on 1000 grain weight and volume weight, while the effect of N top-dressing rate was observed on the crude protein, wet gluten and starch content, and Zeleny index.

**Keywords:** winter wheat; yield; grain quality; leaf diseases; fungicides; nitrogen top-dressing

## 1. Introduction

Mainly bread wheat (*Triticum aestivum*) is grown all over the world, including Europe and Latvia, and the largest bread wheat-growing area is under winter wheat (20 656.14 thousand ha in Europe in 2021, https://ec.europa.eu/ (accessed on 18 November 2022); 448.7 thousand ha in Latvia in 2022, https://stat.gov.lv/ (accessed on 18 November 2022)). The winter wheat yield is variable depending on the country (climate effect) and annual meteorological conditions, e.g., in the European Union in 2022, the average winter wheat yield (bread wheat and spelt wheat (*T. spelta*) yields are reported both together by the Eurostat database) varied from 1.83 t ha$^{-1}$ to 10.87 t ha$^{-1}$ depending on the country. In Latvia in 2022, the average winter wheat yield was 4.36 t ha$^{-1}$, and exceeded 5.0 t ha$^{-1}$ in some very favourable years, reaching even 7.0–9.0 t ha$^{-1}$ in the best farms. The provision of all agrometeorological factors (water, photosynthetically active radiation, nutrients, soil conditions, suitable cultivar, etc.) as well as disease control is equally important for obtaining high winter wheat yields with good grain quality. Several leaf and ear diseases have been recognized as an important wheat-yield-limiting factor globally: tan spot (caused by *Pyrenophora tritici-repentis*), e.g., [1–4]; Septoria leaf blotch (caused by *Zymoseptoria tritici*), e.g., [2,4–6]; Septoria nodorum blotch (caused by *Parastagonospora nodorum*), e.g., [7]; powdery mildew (caused by *Blumeria graminis*), e.g., [4,8]; leaf rust

(caused by *Puccinia recondita*; previously *Puccinia triticina*), e.g., [8]; stripe rust (caused by *P. striiformis*), e.g., [4]; and Fusarium head blight (caused by *Fusarium* spp.), e.g., [6]. Which disease will spread more widely and will be more harmful depends on the specific site and year conditions. In Latvia, tan spot and Septoria leaf blotch have been recognized as the most important wheat leaf diseases [9]. In addition to yield loss, diseases can also cause grain quality decrease [10]. The pressing target among farmers in Latvia is not whether to use a fungicide for winter wheat, but real time application (when and how often) has been of great importance. Nevertheless, after the data analysis of 350 field trials in Sweden, it was found that the use of fungicide was profitable in 188 cases, but was not profitable in 162 cases [11]. It is fairly often concluded that the application of fungicides might not be recommended at all in dry years [5,8], that it is more effective in years with sufficient water provision [4], and that one spraying can provide similar efficacy compared with two or three applications [6,8,12]. These different effects have been obtained in diverse climatic conditions when different diseases prevailed, different fungicides were applied, different cultivars were used, and also the rates of fertilization with nitrogen (N) were different. Several research studies show a strong effect of the fungicide application (F) × nitrogen fertilization rate (N) interaction on grain yield, when higher yields were obtained using the most intensive fungicide application strategy together with the highest N rates [1–3,13]. However, there are also studies in which N × F interaction effect on grain yield has not been established [14], and even in Brinkman et al's [13] study, it did not appear in one out of the nine locations. The literature also shows that the chemical indicators of wheat grain quality were not affected by the F × N interaction [15].

It is observed that producers rarely make fungicide application decisions based solely on expected yield or yield quality losses due to disease outbreaks. Often, the decision is related to the farmer's attitude towards risk, the use of pesticides in general, the farm's financial situation, and other reasons. In addition, no farm has an unsprayed control variant, and if production results show a profit, the unnecessary fungicide application goes unnoticed even financially. However, the fact that fungicides should have been sprayed, but were not, is often obvious [11]. In the most important wheat growing region of Latvia, farmers grow winter wheat comparatively intensively, using high N top-dressing rates and fungicides. For a fungicide application or N top-dressing to be cost-effective, it must pay for itself in increased yield and/or quality. Despite the fact that the impact of nitrogen top-dressing and the application of different fungicides at different timings has been studied thoroughly in the world, the results obtained are contradictory and more research in particular conditions is required. As tan spot and Septoria leaf blotch are the most common wheat diseases in the humid and cool climate [9] of Northern Europe, the triazole (DeMethylation Inhibitor (DMI)) with high efficacy against both diseases was selected in our study—prothioconazole, which was supplemented with spiroxamine (amine, Sterol Biosynthesis Inhibitor (SBI)) in T1 (growth stage (GS) 32–33) treatment and with two active substances from carboxamide group (Succinate Dehydrogenase Inhibitors (SDHI)) in T2 (GS 55–59) treatment; metconazole (DMI) was used in T3 (ear) (GS 63–65) treatment. Therefore, the main objective of this study was to determine the effect of different intensities of fungicide application and N top-dressing rates on winter wheat yield and grain quality. Another objective was to test the hypothesis that an increase in N fertilization rates requires an intensified winter wheat disease control.

Since three trial-years were characterised by shorter or longer periods of drought and heat and a sufficient supply of water was observed only in one year, the hypothesis failed to be proven, but the average four-year yield was significantly affected by both investigated factors. On the other hand, the physical indicators of grain quality were significantly affected by fungicide application, while chemical indicators were significantly affected by increased N top-dressing rates.

## 2. Materials and Methods

### 2.1. Trial Site and Studied Factors

Two-factor field experiments with winter wheat (*Triticum aestivum*) were carried out at the Research and Study farm "Peterlauki" ($56°$ $31'$ N; $23°$ $42'$ E) of the Latvia University of Life Sciences and Technologies for four years (2017/2018–2020/2021). Soil at the site was Epiabruptic Endostagnic Endoprotocalcic Luvisol in the 2017/2018 and 2019/2020, and Cambic Calcisol in the 2018/2019 and 2020/2021 [16]. Soil reaction ($pH_{KCL}$) was 6.4–7.0, the content of $P_2O_5$ was 118–181 mg $kg^{-1}$, the content of $K_2O$ was 122–262 mg $kg^{-1}$, and soil organic matter content varied from 29 to 42 g $kg^{-1}$ depending on the year (agrochemical data refer to the 0–20 cm soil depth). Winter wheat cultivar 'Skagen' (DE) was used. 'Skagen' is widely grown in Latvia and is characterised by a high baking quality (elite group) and a comparatively low susceptibility to common leaf diseases (https://www.bundessortenamt.de/bsa/sorten/beschreibende-sortenlisten/ (accessed on 30 December 2022)). Plot size was 20 $m^2$ (2 m × 10 m), and treatments were arranged randomly in four replications.

Studied factors were as follows: (A) fungicide application (F, five treatments), and (B) nitrogen top-dressing rate (N, four treatments). In total, 20 variants were studied.

Fungicide application variants:

F0—control, without fungicide application;

F1—half of a full fungicide dose sprayed at GS 55–59 (T2);

F2—a full fungicide dose sprayed at GS 55–59 (T2);

F3—a full fungicide dose split in two treatments: at GS 32–33 (T1), and at GS 55–59 (T2);

F4—two full fungicide doses split in three treatments: half—at GS 32–33 (T1), half—at GS 55–59 (T2), and full—at GS 63–65.

In this study, the full fungicide dose (100%) was taken as the fungicide dose according to the highest recorded dose of the triazole active substance prothioconazole (DMI) per hectare (200 g $ha^{-1}$) in one treatment. Prothioconazole was selected due to its efficacy against the most spread wheat leaf diseases in Latvia. In the first treatment (T1), protioconazole was applied at 50% of the full dose (100 g $ha^{-1}$) in combination with the active ingredient spiroxamine (SBI) (187.5 g $ha^{-1}$), which is intended to control powdery mildew in cereals. In the second treatment (T2), a fungicide, which, in addition to prothioconazole (a half or a full dose according to the scheme), contains the active substances of the carboxamide group (SDHI)—bixafen and fluopyram (both—48.75 g $ha^{-1}$ in F1, and 97.5 g $ha^{-1}$ in F2). In the third treatment (T3), a full dose of a fungicide containing the active substance metconazole (DMI) (90 g $ha^{-1}$) was used against Fusarium head blight.

Nitrogen top-dressing variants which were applied:

N120 kg $ha^{-1}$, divided into two portions 80 + 40 kg $ha^{-1}$;

N150 kg $ha^{-1}$, divided into two portions 80 + 70 kg $ha^{-1}$;

N180 kg $ha^{-1}$, divided into three portions 80 + 70 + 30 kg $ha^{-1}$;

N210 kg $ha^{-1}$, divided into three portions 80 + 80 + 50 kg $ha^{-1}$ (further in the text N120, N150, N180, and N210).

The first portion was given at the time of vegetation renewal in spring, the second portion—at GS 31–32, and the third portion—at GS 49–51. Ammonium nitrate (N 34.4%) was used for the first and third portion of top-dressing, and ammonium sulphate (N 21% and S 24% to provide 28.8 kg $ha^{-1}$ of S) and ammonium nitrate were used for the second portion of top-dressing.

### 2.2. Crop Management

The agrotechnology used in the trial was typical for the region in production conditions. The pre-crop was always wheat. Traditional soil tillage including ploughing at the depth of 22 cm was used. The rate of basic fertilizer was calculated with the aim to obtain an 8 t $ha^{-1}$ grain yield, and it was given before sowing: 11–25 kg $ha^{-1}$ N, 33–66 kg $ha^{-1}$ $P_2O_5$, and $K_2O$ depending on the year. Sowing was performed at the optimal time for local conditions (13–27 September depending on the year), and seeds treated with fungicides were used at the rate of 450 (in 2018–2020) to 500 (in 2017) germinable grain $m^{-2}$. For

crop care, herbicides for weed control and plant growth regulators (twice) were applied every year, and insecticides were applied according to the need in 2020 and 2021. Used plant growth regulators were as follows: Cycocel 750 SC (chlormequat chloride, 750 g L$^{-1}$) 1 L ha$^{-1}$ at the GS 28–29 and Medax Top SC (calcium prohexadione, 50 g L$^{-1}$, mepiquat chloride, 300 g L$^{-1}$) 0.75 L ha$^{-1}$ at the GS 33–34. The foliar fertilizer YaraVita Gramitrel (Yara International ASA, Oslo, Norway) (N 3.9%, MgO 15.2%, Cu 3.0%, Mn 9.1%, and Zn 4.1%) was applied in spring together with the plant growth regulator. Grain yield was harvested at GS 89–90, using direct combining (25–26 July 2018, 2019, and 2021, and 2 August 2020), and the yield was recalculated at the 100% purity and 14% moisture.

### 2.3. Observations and Records Made in the Trial

Crop growth stages (GS 11, 21, 31, 32, 33, 37–39, 49, 51, 55, 59, 61–63, 69, 71, 81, 89–90) according to BBCH scale [17] were recorded every year.

Disease severity (%) was recorded visually five times per season: the whole plant was evaluated at the end of tillering–early stem elongation; three upper leaves were evaluated at flag leaf stage and during heading; two upper leaves were evaluated during the milk stage of maturity. The AUDPC (area under the disease progress curve) was calculated to assess the disease impact during the whole vegetation season [18]. Symptoms for every disease were evaluated separately. Fifty plants/leaves/ears were taken per every plot for evaluation, and leaves were taken to keep the proportion of all levels (flag leaf, 2nd leaf, and 3rd leaf). Wheat ears were evaluated at the early milk maturity stage (data not shown). In addition, the leaf green area (LGA) was recorded visually (in %) at the late milk maturity (GS 77) on the upper two leaves.

In 2020, lodging was observed, which was evaluated using a point scale (9–1, where 9 means no lodging, and 1 means very strong lodging and all stems are bent down at a 90° angle).

Before harvesting, two sample sheaves (each from 0.1 m$^2$) were taken from every plot to determine the grain/straw ratio, which was later used to calculate the straw yield from the grain yield.

A grain sample of 1 kg was taken from each plot during harvesting to detect grain quality parameters. The Near Infrared Spectroscopy (NIRS) method (analyser Infratec$^{TM}$ NOVA (FOSS, Hillerød, Denmark)) was used to determine the content of crude protein (CP, % in dry matter), wet gluten (WG, for grain at 14% moisture), and starch (SC, % in dry matter), as well as Zeleny index (ZI) and volume weight (VW, kg hL$^{-1}$). The thousand grain weight (TGW, g) was determined according to the standard ISO 520:2010.

### 2.4. Data Statistical Processing

The analysis of variance (ANOVA) was used for data analysis both in every specific year and taking into account the data of all four years. Differences between treatments were considered as significant at $p \leq 0.05$ and were detected using Bonferroni test. Significantly different values in tables and figures are labeled with different letters in superscript. Correlation and regression analyses were used for discovering the relations between studied parameters. Data were analysed using IBM Statistics for Windows, Version 23.0.

### 2.5. Meteorological Conditions during the Study Period

Meteorological conditions were diverse during trial years (Table 1). In autumn (September, October), the best temperature and moisture conditions for good stand establishment were observed in 2019. The autumn of 2017 was overly wet, but those of 2018 and 2020—overly dry. In general, conditions were good for wintering in all four winters, except early spring of 2021, when snow mold (caused by *Microdochium nivale* and/or *Typula* spp.) was observed. The whole spring and summer period of 2018 was extremely hot and dry, but that of 2019 and 2021 was characterised with several drought periods. Only the year 2020 was suitable for the formation of high winter wheat yields (Figure 1).

**Table 1.** Data of mean air temperatures and precipitation in trial period (2017/2018–2020/2021) and in comparison with the long-term-average data, Research and Study farm "Peterlauki", Latvia University of Life Sciences and Technologies.

| Month | Average Air Temperature, °C | | | | Norm, °C | Precipitation, mm | | | | Norm, mm |
|---|---|---|---|---|---|---|---|---|---|---|
| | 2017/2018 | 2018/2019 | 2019/2020 | 2020/2021 | | 2017/2018 | 2018/2019 | 2019/2020 | 2020/2021 | |
| Sept. | 13.0 | 14.9 | 12.7 | 14.9 | 12.3 | 79.8 | 25.5 | 53.6 | 16.2 | 59.9 |
| Oct. | 6.9 | 8.5 | 9.0 | 9.8 | 6.9 | 80.0 | 10.6 | 36.4 | 58.4 | 68.2 |
| Nov. | 3.9 | 3.0 | 4.4 | 5.7 | 2.5 | 45.4 | 6.8 | 48.4 | 12.6 | 50.4 |
| Dec. | 1.3 | −0.9 | 2.6 | 0.5 | −0.9 | × | × | × | × | 47.1 |
| Jan. | −1.2 | −4.2 | 3.2 | −3.4 | −2.7 | × | × | × | × | 43.6 |
| Febr. | −6.8 | 1.2 | 2.5 | −5.7 | −2.7 | × | × | × | × | 34.8 |
| Mar. | −2.0 | 3.0 | 3.1 | 1.9 | 0.7 | × | 29.6 | 27.0 | 13.6 | 33.8 |
| Apr. | 9.0 | 8.1 | 6.1 | 5.9 | 6.7 | 69.5 | 0.0 | 9.2 | 4.7 | 36.0 |
| May | 16.1 | 12.4 | 9.9 | 11.1 | 12 | 12.0 | 20.4 | 30.2 | 50.6 | 52.4 |
| June | 16.8 | 19.4 | 18.7 | 19.2 | 15.5 | 16.0 | 8.6 | 139.6 | 14.8 | 73.4 |
| July | 20.7 | 16.8 | 17.0 | 22.0 | 17.9 | 56.5 | 101.0 | 47.7 | 3.2 | 82.1 |
| Aug. | 19.4 | 17.6 | 17.7 | No data | 17.0 | 34.0 | 37.8 | 65.0 | No data | 69.4 |

Meteorological data were registered at the trial site by an automatic weather observation station; the norm means long-term-average data (last 30 years), which were taken from the closest meteorological station (in Jelgava) of the Latvian Environment, Geology and Meteorology Centre (https://videscentrs.lvgmc.lv/ (accessed on 5 September 2022)); ×—precipitation data during winter are not shown, because they can be imprecise due to precipitation in the form of snow in those months.

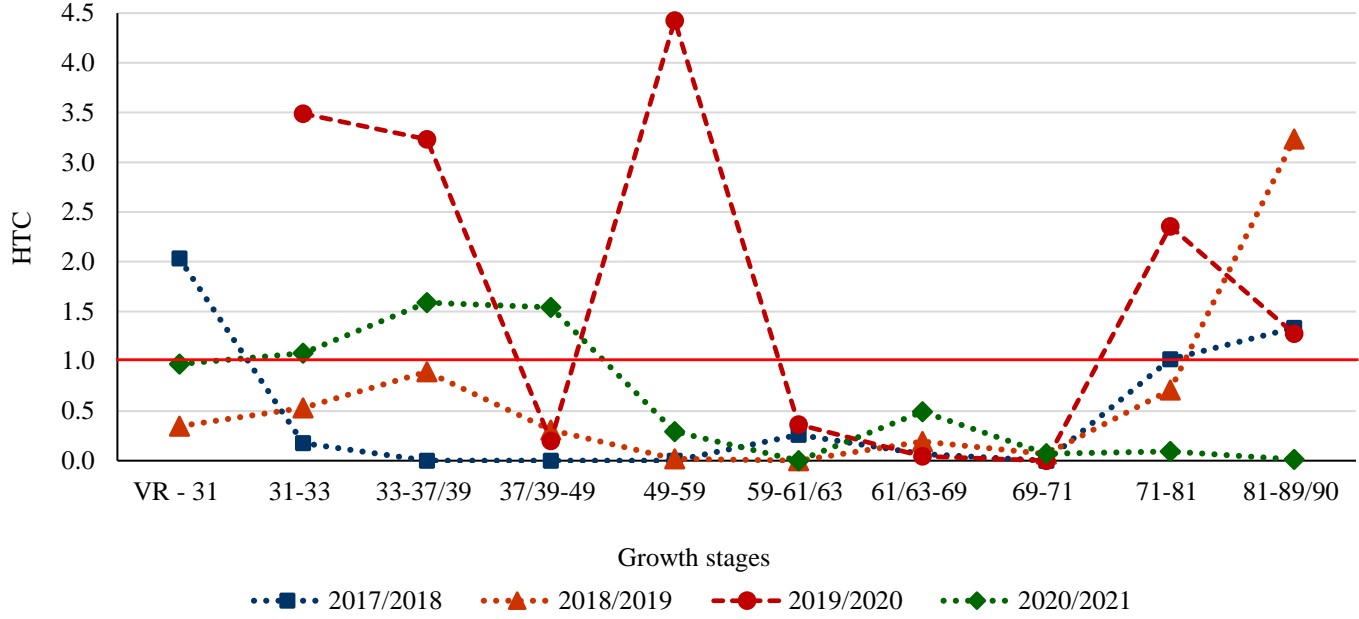

**Figure 1.** Hydrothermal coefficient (HTC) during various growth stages of winter wheat in 2017/2018–2020/2021 at the Research and Study farm "Peterlauki, Latvia University of Life Sciences and Technologies (VR—vegetation renewal; GS—growth stage; HTC was not calculated for the stage VR–GS 31 in 2019/2020, when the average day-and-night temperature only once per 21 days exceeded 10 °C).

In order to characterise the temperature and moisture conditions more accurately in different wheat development stages in the spring-summer vegetation period, the Selyaninov's hydrothermal coefficient (HTC) [19] was calculated according to the Formula (1):

$$\text{HTC} = \frac{\sum R \times 10}{\sum t} \qquad (1)$$

$\sum R$—sum of precipitation per period, which has to be characterised;
$\sum t$—sum of temperatures above 10 °C per the same period.

We used the following criteria for the interpretation of HTC: >2—overly wet; 1–2—sufficient moisture provision; <1—insufficient moisture provision; 1.0–0.7—dry conditions; 0.7–0.4—very dry conditions.

Temperature and precipitation in different growth stages caused a critical lack of water, which was observed in the whole season of 2017/2018 and 2018/2019, as well as from the GS 49 to the harvesting maturity in 2020/2021 (HTC below 1 in Figure 1).

A sufficient supply of water during ear formation (up to GS 32) was noted in 2020 and 2021, and up to GS 31—also in 2018. Grain filling stage (see GS 71–81 in Figure 1) was best provided with moisture in the year 2020, when also temperature was more moderate, while the worst situation was observed in 2021.

## 3. Results

### 3.1. Crop and Disease Development

The lengths of the spring–summer vegetation period, from the vegetation renewal to GS 89–90 (harvesting maturity), differed depending on the trial year: the longest—in 2019/2020 (120 days); the shortest—in 2017/2018 (104 days). The lengths of specific growth stages (GS) were also diverse depending mainly on the temperature and moisture conditions in a specific year, but did not depend on the studied factors—fungicide application or N top-dressing rate.

Severe lodging (2.2–3.1 points) was observed only once over all four years—in 2020 (data are not shown), when a storm with heavy rain was noted on 29–30 June at the early milk stage (GS 71 was noted on 27 June) and lodging remained until harvesting. The rating did not depend on fungicide treatment but was significantly affected by N top-dressing rate; although, a significantly ($p = 0.02$) lower average rating was noted only for the variant N210. Lodging affected the values of TGW and VW (see below the Section 3.3. Winter wheat grain quality).

The development of diseases differed significantly depending on the year ($p < 0.001$). Tan spot (caused by *P. tritici-repentis*) dominated in three years out of the four, achieving the highest level in 2019. Septoria leaf blotch (caused by *Z. tritici*) proved to be the most important wheat disease only in 2020. All other leaf and ear diseases were observed only occasionally, and their severity did not reach 1% in the untreated variant. Fungicide treatment significantly decreased the severity of tan spot and Septoria leaf blotch ($p < 0.001$), but the efficacy of application schemes depended on the year (Table 2). The influence of nitrogen top-dressing rate on the development of diseases was not significant ($p > 0.05$) on average per four years and in each particular trial year as well.

**Table 2.** Development of winter wheat leaf diseases depending on fungicide treatment and year (2018–2021) at the Research and Study Farm "Peterlauki", Latvia University of Life Sciences and Technologies (data in AUDPC units).

| Fungicide Treatment | Tan Spot | | | | | Septoria Leaf Blotch | | | | |
|---|---|---|---|---|---|---|---|---|---|---|
| | 2018 | 2019 | 2020 | 2021 | Average | 2018 | 2019 | 2020 | 2021 | Average |
| F 0 | 12.9 a | 141.9 a | 45.4 a | 61.5 a | 65.4 A | 1.1 a | 2.9 a | 57.3 a | 26.4 a | 21.9 A |
| F 1 | 6.7 b | 90.5 b | 17.8 b | 37.6 b | 38.1 B | 0.9 a | 1.6 b | 21.8 b | 13.2 b | 9.4 B |
| F 2 | 4.1 c | 71.3 c | 16.4 b | 33.6 bc | 31.3 B | 1.3 a | 1.3 b | 21.4 b | 13.9 b | 9.5 B |
| F 3 | 3.9 c | 60.7 cd | 10.5 b | 34.5 bc | 27.4 B | 0.7 a | 0.8 a | 12.9 c | 10.2 b | 6.2 B |
| F 4 | 3.2 c | 45.6 d | 12.7 b | 27.1 c | 22.1 B | 1.3 a | 0.8 b | 16.9 bc | 9.2 b | 7.1 B |

F0—control without fungicide application; F1—half dose applied as T2; F2—full dose applied as T2; F3—full dose applied as split spraying: T1 and T2; F4—two full doses applied as split spraying: T1, T2, and T3. T1—spraying at GS 32–33; T2—spraying at GS 55–59; T3—spraying at GS 63–65. Different letters mean significant differences between disease level (expressed as AUDPC units) for each year (a, b, c, d) and on average for a treatment (A, B).

The efficacy of fungicide application schemes was influenced by disease pressure, and in 2019 (the highest severity of tan spot), a more intensive application of fungicides gave better disease control results. Although the half dose of fungicides gave a lower efficacy

in the majority of cases, on average during the four years the differences in disease level between fungicide application variants were not significant.

Leaf green area (LGA) during the late milk ripening (GS 77) fluctuated on average between 50% and 87% in untreated variants and between 62% and 91% in variants with fungicide application. Leaf green area was significantly influenced by the trial year ($p < 0.001$). Fungicide treatment increased the LGA ($p < 0.001$) on average, although the differences between variants with different fungicide application intensities were significant but small. The effect of nitrogen top-dressing rate on LGA was not clearly established in particular years, but on average per trial period a higher dose of nitrogen slightly and significantly increased the LGA (Figure 2).

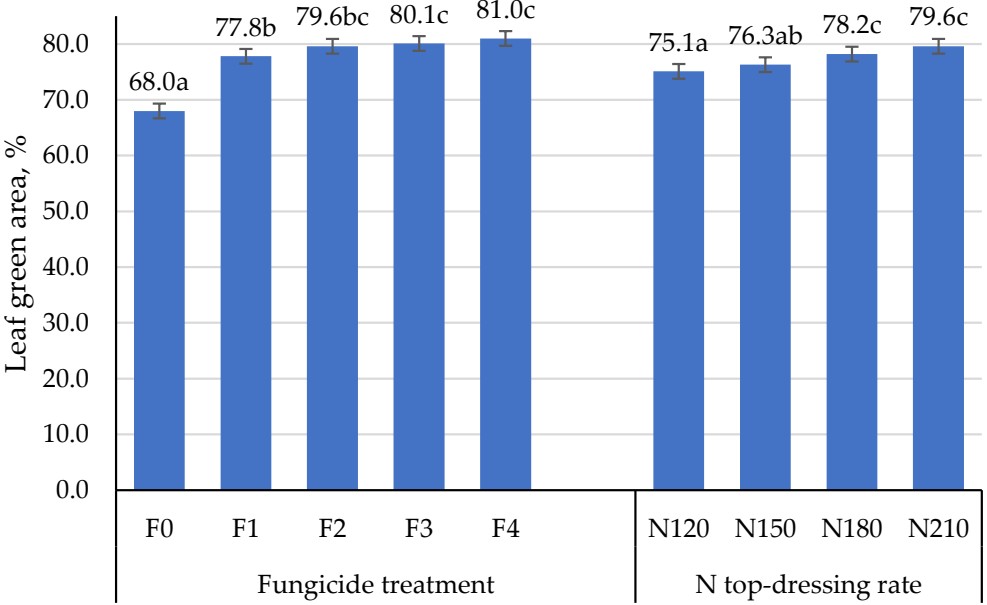

**Figure 2.** Mean four-year (2018–2021) leaf green area (LGA) of winter wheat at late milk stage (GS 77) at the Research and Study farm "Peterlauki", Latvia University of Life Sciences and Technologies depending on fungicide treatment and N top-dressing rate (F0—control without fungicide application; F1—half dose applied as T2; F2—full dose applied as T2; F3—full dose applied as split spraying: T1 and T2; F4—two full doses applied as split spraying: T1, T2, and T3. T1—spraying at GS 32–33; T2—spraying at GS 55–59; T3—spraying at GS 63–65. N120–210—N top-dressing rate in kg ha$^{-1}$ pure N. Different letters mean significant differences between LGA depending on the specific factor).

*3.2. Winter Wheat Grain and Straw Yield*

The average four-year winter wheat grain yield was 6.79 t ha$^{-1}$, which did not reach the planned 8 t ha$^{-1}$. The average yield was affected significantly by fungicide treatment, N top-dressing, and the interaction of both, but the factor influencing the yield most was the trial year (Table 3).

Any fungicide treatment produced a small but statistically significant increase in the average four-year grain yield compared to the control. However, differences between sprayed variants were inconclusive, and, e.g., the average grain yield in variant F1, where half of the fungicide dose was applied, was equivalent to the yield in variant F4, where two full doses of the fungicide in three sprayings were applied (Table 4).

Moreover, the analysis of variance showed a small but significant effect of F × N interaction on the mean four-year wheat yield (Table 3). Although the effect of F × N interaction was significant, clear regularities that the use of higher N top-dressing rates requires more intensive spraying with fungicides, failed to be proved by our results on average per trial period as well as in separate years. For example, the highest mean yield was obtained in variant F4 (two full fungicide doses split in three sprays), when N210 (the highest rate) was used. Moreover, the rate N180 is high, but its use provided

the best yield in fungicide treatment F1 (half dose, one spraying) (Table 4). Such results could be connected with the high influence of the trial year (Table 3, Figure 3) due to the diverse meteorological conditions, especially the differing water supply (Table 1, Figure 1). Fungicide application resulted in a significant wheat grain yield increase only in one trial year (2020); in two years (2019 and 2021) it did not cause significant yield changes, whereas in 2018, the application of fungicides caused a significant negative effect, i.e., the yield decreased (Table 3, Figure 3).

**Table 3.** Mean squares of winter wheat yield under five fungicide treatments, four N top-dressing rates in four years (2018–2021), and five fungicide treatments and four N rates in every specific year, Research and Study farm "Peterlauki", Latvia University of Life Sciences and Technologies.

| Source of Variation | Four-Year Data | | Separate Year Data | | | | |
|---|---|---|---|---|---|---|---|
| | df | Mean Squares | df | 2017/2018 | 2018/2019 | 2019/2020 | 2020/2021 |
| Fungicide (F) | 4 | 0.579 *** | 4 | 0.896 *** | 0.137 | 1.955 *** | 0.038 |
| N top-dressing rate (N) | 3 | 1.130 *** | 3 | 0.249 *** | 0.532 * | 0.236 *** | 0.434 *** |
| Year (Y) | 3 | 136.149 *** | – | – | – | – | – |
| F × N | 12 | 0.145 ** | 12 | 0.170 *** | 0.082 | 0.079 *** | 0.116 *** |
| F × Y | 12 | 0.816 *** | – | – | – | – | – |
| Error | 285 | 0.067 | 60 | 0.017 | 0.175 | 0.031 | 0.019 |
| Total | 320 | | 80 | | | | |

Significant at: * $p = 0.05$; ** $p = 0.01$; *** $p < 0.01$.

**Table 4.** Average four-year (2018–2021) winter wheat grain yield (t ha$^{-1}$) depending on fungicide treatment and N top-dressing rate at the Research and Study farm "Peterlauki", Latvia University of Life Sciences and Technologies.

| Fungicide Treatment | N Top-Dressing Rate | | | | Average for Fungicide Treatment |
|---|---|---|---|---|---|
| | N120 | N150 | N180 | N210 | |
| F0 | 6.51 | 6.64 | 6.70 | 6.75 | 6.65 a |
| F1 | 6.79 | 6.92 | 7.01 | 6.84 | 6.89 b |
| F2 | 6.66 | 6.81 | 6.83 | 6.71 | 6.75 c |
| F3 | 6.54 | 6.75 | 6.97 | 6.87 | 6.78 cd |
| F4 | 6.57 | 6.89 | 6.90 | 7.10 | 6.86 bd |
| Average for N top-dressing | 6.62 a | 6.80 b | 6.88 c | 6.85 bc | × |

F0—control without fungicide application; F1—half dose applied as T2; F2—full dose applied as T2; F3—full dose applied as split spraying: T1 and T2; F4—two full doses applied as split spraying: T1, T2, and T3. T1—spraying at GS 32–33; T2—spraying at GS 55–59; T3—spraying at GS 63–65. N120–210—N top-dressing rate in kg ha$^{-1}$ pure N. Different letters mean significant differences between average yields depending on the studied factor.

Despite the fact that fertilization in general and N top-dressing did not ensure the planned winter wheat yields in three (2018, 2019, and 2021) out of the four years, the increase in N rate always caused a significant increase in grain yield (Tables 3 and 4, Figure 3). Grain yield increased up to the rate of N150–N180 depending on the year.

The mean four-year wheat straw yield was significantly affected only by the trial year ($p < 0.01$), varying from 5.98 to 12.08 t ha$^{-1}$ depending on the year (Figure 4). The exception was 2018, when the yield of straw was significantly ($p = 0.008$) affected by fungicide application, which caused a decrease in straw yields in variants F3 and F4 (data not shown). The straw yield is important in the formation of total wheat biomass, which also requires water and nutrition elements, including part of N top-dressing rate.

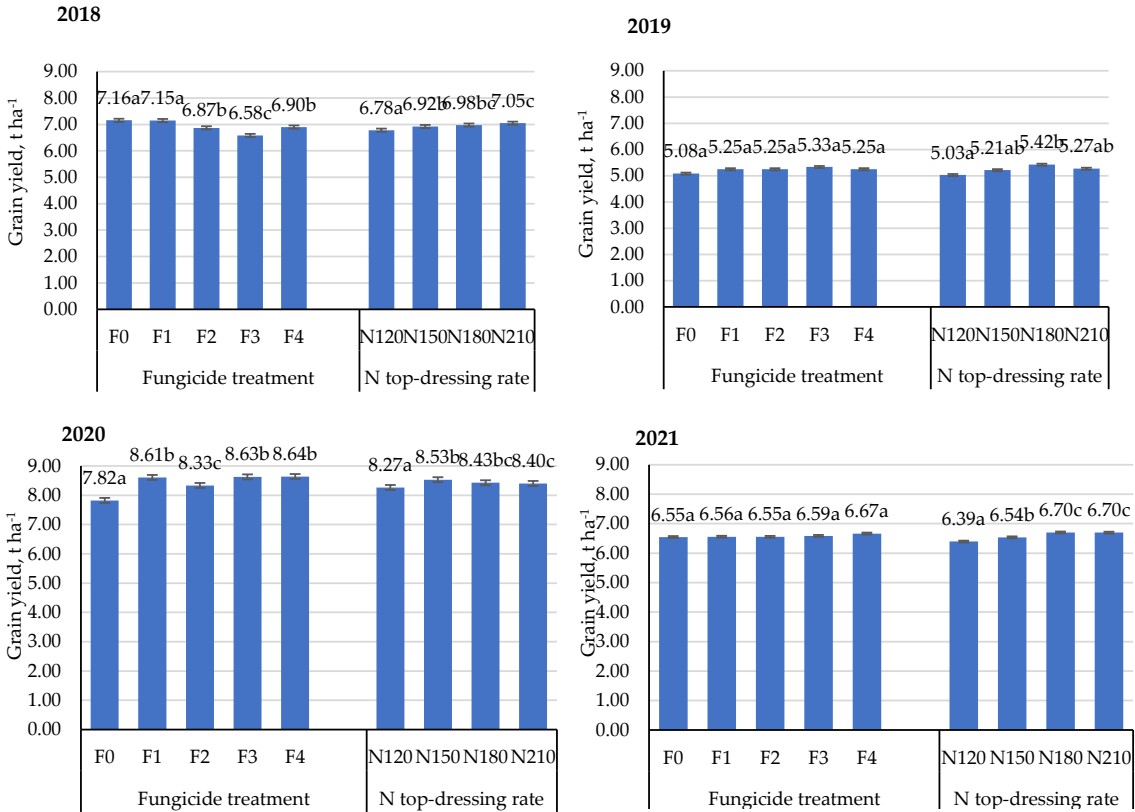

**Figure 3.** Winter wheat grain yield depending on fungicide treatment, N top-dressing rate, and trial year (2018–2021) at the Research and Study farm "Peterlauki", Latvia University of Life Sciences and Technologies (F0—control without fungicide application; F1—half dose applied as T2; F2—full dose applied as T2; F3—full dose applied as split spraying: T1 and T2; F4—two full doses applied as split spraying: T1, T2, and T3. T1—spraying at GS 32–33; T2—spraying at GS 55–59; T3—spraying at GS 63–65. N120–210—N top-dressing rate in kg ha$^{-1}$ pure N. Different letters mean significant differences between yields depending on the specific factor in the specific year).

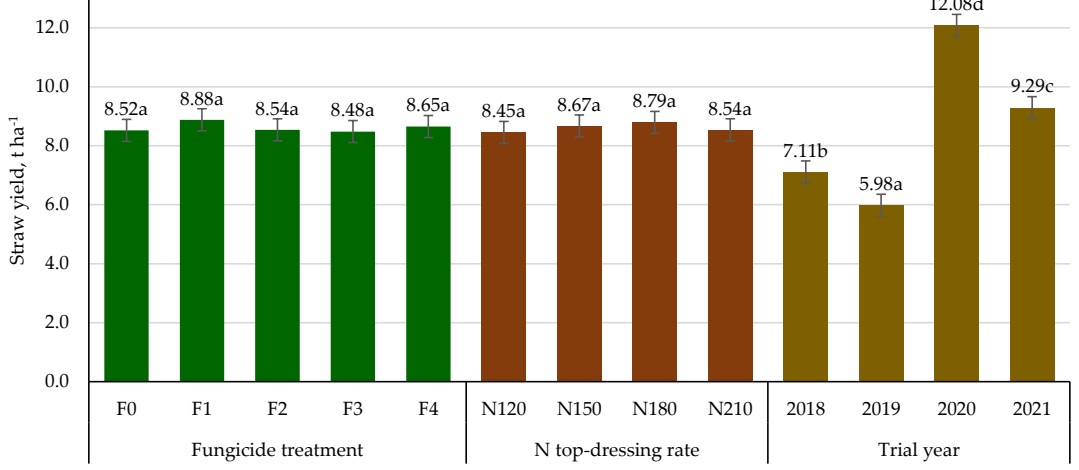

**Figure 4.** Winter wheat straw yield depending on fungicide treatment, N top-dressing rate, and trial year (2018–2021) at the Research and Study farm "Peterlauki", Latvia University of Life Sciences and Technologies (F0—control without fungicide application; F1—half dose applied as T2; F2—full dose applied as T2; F3—full dose applied as split spraying: T1 and T2; F4—two full doses applied as split spraying: T1, T2, and T3. T1—spraying at GS 32–33; T2—spraying at GS 55–59; T3—spraying at GS 63–65. N120–210—N top-dressing rate in kg ha$^{-1}$ pure N. Different letters mean significant differences between yields depending on the specific factor).

### 3.3. Winter Wheat Grain Quality

3.3.1. Physical Grain Quality Indicators—1000 Grain Weight (TGW) and Volume Weight (VW)

Four-year average values of TGW and VW significantly depended on fungicide treatment ($p < 0.01$ and $p = 0.046$, respectively) and especially strongly on the trial year ($p < 0.01$ for both indicators; see Table 5). Although TGW increased in three years of fungicide use (2019, 2020, and 2021), this increase was significant only in 2019/2020 (Table 5). VW increased significantly in three trial years except 2020/2021. Lower TGW and VW values were noted in 2020 as a result of heavy lodging, and in 2021—as a result of drought during grain filling (Figure 1). In years when a significant TGW and VW increase was observed after fungicide application, differences between treated variants were noted only once: for VW when treatment F4 did not cause an increase in VW if compared with the control (F0).

**Table 5.** Winter wheat TGW and VW in trial years (2017/2018–2020/2021) at the Research and Study farm "Peterlauki", Latvia University of Life Sciences and Technologies, and the effect of studied factors on their values.

| Trial Year | Average (Min–Max) | Effect of Studied Factors: *p*-Value | | |
|---|---|---|---|---|
| | | F | N | F × N |
| *1000 grain weight (TGW), g* | | | | |
| 2017/2018 | 46.29 b (44.81–47.58) | 0.289 | 0.160 | 0.366 |
| 2018/2019 | 49.20 a (47.74–50.35) | 0.158 | 0.204 | 0.988 |
| 2019/2020 | 43.99 c (42.16–45.99) | 0.001 | 0.420 | 0.653 |
| 2020/2021 | 36.06 d (34.04–37.31) | 0.266 | 0.290 | 0.724 |
| *Volume weight (VW), kg hL$^{-1}$* | | | | |
| 2017/2018 | 82.73 a (81.77–83.23) | 0.006 | 0.061 | 0.738 |
| 2018/2019 | 79.72 b (79.27–80.22) | 0.001 | 0.687 | 0.013 |
| 2019/2020 | 77.92 c (76.72–78.58) | 0.001 | 0.213 | 0.900 |
| 2020/2021 | 72.56 d (70.08–73.39) | 0.880 | 0.509 | 0.875 |

F—fungicide treatment; N—nitrogen top-dressing. Different letters mean significant differences between TGW and VW values depending on trial year.

The relationship between TGW and the important indicators leaf green area at the late milk stage (LGA), grain yield, VW, and crude protein (CP) content in grain was evaluated. The preservation of LGA during milk ripeness stage is important because of grain filling and TGW formation at that time. Frequently, a higher TGW is connected with a higher VW, which was also observed in our study in 2018 and 2019, when a higher TGW and a significantly higher VW were noted. At the same time, TGW is not only a physical grain quality indicator but also a yield-forming component. In all trial years, no significant TGW relationship with any of mentioned indicators was established. The correlation of TGW with LGA and grain yield was significant in three years, and with VW and CP—in two years (Table 6).

**Table 6.** Correlation of TGW with LGA at the end of milk ripeness, grain yield, VW, and CP depending on trial year (2017/2018–2020/2021), Research and Study farm "Peterlauki", Latvia University of Life Sciences and Technologies, ($n = 20$).

| Trial Year | Correlation of TGW with Other Indicators: Correlation Coefficients | | | |
|---|---|---|---|---|
| | LGA | Grain Yield | VW | CP |
| 2017/2018 | NS | 0.688 ** | 0.639 ** | 0.639 ** |
| 2018/2019 | 0.592 ** | 0.478 * | NS | 0.655 ** |
| 2019/2020 | 0.765 ** | 0.576 ** | 0.899 ** | NS |
| 2020/2021 | 0.445 * | NS | NS | NS |

TGW—1000 grain weight; LGA—leaf green area at the end of milk ripeness; VW—volume weight; CP—crude protein. NS means $p > 0.05$; * $p = 0.05$; ** $p = 0.01$.

A significant correlation between LGA and both TGW and VW, and between TGW and VW was also found when data of all four years ($n = 80$) were included in correlation analysis ($p = 0.01$; data are not shown).

3.3.2. Chemical Grain Quality Indicators—Crude Protein (CP) and Wet Gluten (WG) Content, Zeleny Index (ZI), and Starch Content (SC)

Unlike the physical quality indicators, the mean four-year values of CP, WG, ZI, and SC were not affected significantly ($p > 0.05$) by fungicide treatment variants, but they depended significantly on N top-dressing rate and trial year (Table 7).

**Table 7.** Winter wheat CP, WG, ZI, and SC (measured by NIRS) in trial years (2017/2018–2020/2021), and the effect of studied factors on their values, Research and Studies farm "Peterlauki", Latvia University of Life Sciences and Technologies.

| Trial Year | Average (Min–Max) | Effect of Studied Factors: *p*-Value | | |
|---|---|---|---|---|
| | | F | N | F × N |
| *Crude protein (CP), % in dry matter* | | | | |
| 2017/2018 | 11.4 a (10.8–11.9) | 0.002 | 0.001 | 0.889 |
| 2018/2019 | 13.7 b (13.0–14.2) | 0.730 | 0.001 | 0.622 |
| 2019/2020 | 13.9 c (13.3–14.5) | 0.666 | 0.001 | 0.855 |
| 2020/2021 | 13.8 bc (12.0–15.3) | 0.886 | 0.001 | 0.955 |
| *Wet gluten content (WG), % in grain with 14% moisture* | | | | |
| 2017/2018 | 22.8 a (20.3–24.3) | <0.001 | <0.001 | 0.836 |
| 2018/2019 | 29.1 b (26.9–30.6) | 0.481 | <0.001 | 0.626 |
| 2019/2020 | 29.7 b (28.3–31.0) | 0.829 | <0.001 | 0.843 |
| 2020/2021 | 28.4 b (22.7–32.9) | 0.879 | <0.001 | 0.960 |
| *Zeleny index (ZI) for grain with 14% moisture* | | | | |
| 2017/2018 | 31.9 a (26.0–35.2) | 0.001 | <0.001 | 0.961 |
| 2018/2019 | 50.5 b (45.1–54.8) | 0.309 | <0.001 | 0.617 |
| 2019/2020 | 52.8 d (48.7–56.7) | 0.826 | <0.001 | 0.871 |
| 2020/2021 | 51.3 c (36.2–62.7) | 0.820 | <0.001 | 0.921 |
| *Starch content (SC), % in dry matter* | | | | |
| 2017/2018 | 69.7 a (69.2–70.2) | <0.001 | <0.001 | 0.059 |
| 2018/2019 | 68.1 b (67.5–69.1) | 0.999 | <0.001 | 0.765 |
| 2019/2020 | 67.1 c (66.1–68.0) | 0.165 | <0.001 | 0.943 |
| 2020/2021 | 65.6 d (63.7–67.9) | 0.966 | <0.001 | 0.984 |

F—fungicide treatment; N—nitrogen top-dressing. Different letters mean significant differences between CP, WG, ZI, and SC values depending on trial year.

The year 2018 was the only trial year when fungicide treatment significantly affected all measured parameters (Table 7) but in the direction not desired by the grower. The use of a full fungicide dose (F2 and F3) and two full doses (F4) caused a significant decrease in the CP and WG content, and in ZI. At the same time, SC increased significantly in variants F1, F2, and F3, but in F4 it was equivalent to that of the variant F0. In all other trial years (2019–2021), the use of fungicide did not significantly affect the chemical wheat grain quality indicators. The interaction between both studied factors (F × N) never affected the CP, WG, ZI, and SC values significantly. The increase in nitrogen top-dressing rate caused a significant increase in CP, WG, and ZI up to the rate of N210 in 2018–2020, and up to the rate of N180 in 2021. The rate of N150 always resulted in an increase in CP, WG, and ZI compared with the variant where N120 was used. According to research findings, CP content often correlates with SC. Also in our trial, we observed a significant negative correlation of CP with SC (correlation coefficients depending on the year in 2018–2021, respectively: $-0.763$, $-0.971$, $-0.972$, and $-0.996$; $n = 20$; $p = 0.01$). Thus, changes in SC depending on N top-dressing rate were opposite to the changes in CP content—the gradual decrease in SC was significant up to N210 in 2018, 2019, and 2021; whereas in 2020,

although SC decreased gradually, the decrease was significant and of equal value only in variants N180 and N210 compared with the variant N120.

## 4. Discussion

Tan spot and Septoria leaf blotch were the main diseases during the trial period, which supports our previous conclusions related to the situation in the Baltic region [9,12]. As it has been found before, wheat as a pre-crop for wheat increases the level of wheat diseases, but ploughing can mitigate this impact significantly. The influence of meteorological conditions on the development of diseases has been recognized as more significant than the impact of agronomic practices [9]. The disease progress curves of leaf blotches in the present study differed compared with other findings—a rapid development of diseases started only at the time of flowering or even later in our trial. The results showed that, for example, in 2019, when tan spot achieved the highest level, its severity was 2% at GS 71, and two weeks later (GS 77) achieved 19% in the untreated variant. Similarly, in 2020, when Septoria leaf blotch dominated, the disease severity increased from 1.5% to 11.1% at the same growth stages (from GS 71 to 77). Those peculiarities of leaf disease development explain the comparatively low efficacy of T1, because that time of spraying did not coincide with the development of leaf diseases. The efficacy of fungicide treatment depends not only on the total pressure of diseases during the season, but also on crucial periods for disease development during the season.

More detailed data related to winter wheat leaf disease development depending on fungicide application and N top-dressing rate have been presented in other articles, e.g., by Švarta et al. [20].

The results revealed that the influence of the meteorological conditions on the yield of winter wheat in the research years was greater than the influence of the studied factors (Table 3, Figures 3 and 4), despite the fact that both of them—fungicide application and N top-dressing rate—significantly affected the average four-year yield. According to the obtained data (Table 4), any variant of fungicide application gave a significant increase in the average grain yield, but the differences in yield among the sprayed variants were inconclusive, e.g., the grain yield in the F1 variant (half of the fungicide dose was applied) was equivalent to the yield in the F4 variant (two full fungicide doses were applied in three sprayings). Obtained data are supported by similar results regarding the development of leaf diseases—fungicide application decreased the severity of diseases, but the increase in fungicide treatment intensity was not effective. Similar results have been also obtained in the previous studies in Latvia, where both the pre-developed fungicide application schemes and the two decision support systems were used in one trial, and the applied fungicides included also strobilurins. The results showed that any variant of fungicide application ensured an increase in the yield, but significant differences between yields in the variants differently treated with fungicides were not established [12]. This conclusion is supported by several studies conducted in different conditions and in different regions. A study in Luxembourg [6] found that a single spraying according to the recommendation of a decision support system can provide a yield equivalent to that provided by two or three sprayings according to a previously developed scheme. In Canada it was concluded that a single spraying at GS 39 provided the yields equivalent to those obtained in the variants where a split spraying was applied twice (at GS 30 and 39) [21]. Previous findings in Latvia [12] as well as other studies demonstrate that the yield increase as a result of fungicide application depends on meteorological conditions [5,21–24] and also on other agrotechnical factors (pre-crop, tillage system, etc.) [25]. A study of Byamukama et al. [26] proved that the positive effect of fungicide application for yield increase is better manifested if there are sufficient moisture conditions at the time most favourable for disease development during the vegetation season. During our research, sufficient water provision in the vegetation period was observed only in 2020, when any variant of fungicide application provided a significant yield increase (Figure 3); the other trial years were marked by drought in the stages important for the formation of winter wheat yield (Figure 1). In two years (2019

and 2021), fungicide treatment did not significantly affect the wheat grain yield (Figure 3), and the yield in the variants with fungicide application (F1–F4) was equivalent to that of the control variant (F0). Similar were the findings of Fernandez et al. [27] who carried out a study of *Triticum durum* in Canada. The researchers established that for disease control, one fungicide spraying (at GS 62–65) was of equal value to two sprayings (at GS 31 or 49 and at GS 62–65), but the yield in all treated variants was numerically but not significantly higher than in the control variant. In 2018, when the shortest vegetation period was observed and the season was extremely hot and dry (Table 1, Figure 1), the effect of fungicide application on the yield was significant but negative, i.e., a significant yield reduction was observed in three variants (F2–F4) (Figure 3). In that year, the disease severity was low and only tan spot was spread more pronouncedly. We hypothesized that the drought stress together with the stress caused by fungicide spraying resulted in a yield reduction in variants F2–F4. Such hypothesis about a combined effect of both stresses was also expressed by Rodrigo et al. [5], who observed that in dry years, all variants of fungicide spraying led to a decrease in grain yield. If decision support systems were used, it would have been possible to avoid such a situation, because in conditions that do not promote the spread of diseases, the system would likely not recommend spraying [8]. The studies comprising simultaneously several cultivars report that cultivars with a lower yield potential are more responsive to the application of fungicide and provide a greater increase in yield (e.g., in Bhatta et al. [23]—fungicide applied at GS 39; in Byamukama et al. [26]—fungicide applied at GS 60), or—the effect of fungicide application depends on the cultivar's genetic characteristics, which is related to the year of the registration of the cultivar [28]. The cultivar 'Skagen' used in our study is characterized with at least medium yield potential and has a relatively good field resistance against the main leaf diseases (3 to 4 points in a 9-point scale, where 9 means the highest susceptibility; https://www.bundessortenamt.de/bsa/sorten/beschreibende-sortenlisten/ (accessed on 30 December 2022)). On the other hand, Morgunov et al. [29], while studying the use of fungicides against leaf rust (caused by *Puccinia recondita*), found an increase in yield as a result of fungicide application for cultivars with different levels of resistance to this disease. In our study, *Puccinia recondita* was observed only once—at milk ripeness in 2021, when the severity of the disease was low.

Although drought did not contribute to N-use efficiency in three of the four study years, our results suggest that in each study year separately (Table 3, Figure 3) and on average over the entire period (Table 4), N top-dressing had a significant effect on wheat grain yield. On average during the study period, a significant increase in yield was noted by increasing the N top-dressing rate to N180; however, the results depended on the specific year, e.g., in 2020, a significant yield increase was noted up to the rate N150 (Figure 3). The results were in line with the previous studies, when wheat yield increased significantly with the increase in N rate up to 120 kg ha$^{-1}$ [30], 153 kg ha$^{-1}$ [31], or 180 kg ha$^{-1}$ [32,33]. In addition, the N-rate up to which the yield increased significantly, was influenced by the agrometeorological conditions of the specific study.

At the start of the study, it was hypothesized that higher N rates would probably produce a denser wheat biomass, which would stimulate more diseases in the crop and therefore require a more intensive use of fungicides. However, in our case, the above-ground biomass-forming component, straw yield, did not increase by increasing the N top-dressing rate, nor did it depend on either F or F $\times$ N; it was influenced only by the conditions of the trial year (Figure 4). In other studies, e.g., [13], it was found that fungicide application strategy at GS 39 or GS 60–65 ensured the highest and a consistent grain yield increase, especially if it was supplemented with high N rates. Our results also revealed a small significant impact of F $\times$ N interaction on the average wheat grain yield over the entire research period and in three separate years (2018, 2020, and 2021; Table 3). However, it was not possible to establish any regularity that in more intensive fertilizing options, also a more intensive spraying should be used (Table 4). The inconsistency of the F $\times$ N impact on yield increase was even more expressed in specific years (data not shown) compared

with the average four-year result. In Argentina, where a significant effect of the F × N interaction on yield was also found in the wheat artificially inoculated with *P. tritici-repentis*, it was established that increasing N rates in a fungicide-untreated control variant did not result in a significant increase in yield, while the yield increase in fungicide-treated (all variants included the use of strobilurins) variants was significant [3]. We observed a significant yield increase by increasing the N top-dressing rate also in untreated control. Our research was performed in a natural background of infection, and the infection level was not high. The results of different studies can differ depending also on cultivar and agrometeorological conditions of a specific study.

The effect of the studied factors on both groups of grain quality indicators—physical (TGW and VW) and chemical (CP, WG, ZI, and SC)—were found to be different. During the four-year trial period, the average physical grain quality indicators were significantly affected by fungicide treatment, but chemical quality indicators—by N top-dressing rate. Both groups were affected significantly by meteorological conditions of the study year, but were not affected by F × N interaction.

The increase in TGW due to fungicide application is established also by other researchers [23,27,34], but Landolfi et al. [34] have also pointed to the importance of the meteorological conditions of the trial year. We found a significant correlation of LGA at the end of milk ripeness stage with TGW, which is in line with MacLean et al. [24], who pointed to the importance of LGA increase for the formation of TGW in the result of fungicide application. The fact that the increase in N top-dressing rate does not significantly increase TGW is consistent with the results of Landolfi et al. [34] but differs from other findings [32,35], where also lower N rates (N60 and N90) and unfertilized control were included. The lowest N top-dressing rate used in our study was N120 kg ha$^{-1}$, which is not that small at all. Some researchers indicate an increase in TGW up to a certain N top-dressing rate which, when exceeded, decreases the TGW value [36]. For producers, VW is even a more important indicator than TGW; therefore, it is used by grain buyers for price determination. VW depended mostly on the meteorological conditions of the year (Table 5): strong lodging in 2020 and the lack of water during part of the 2021 season (Table 1, Figure 1) caused lower values of VW. As already mentioned, fungicide application increased VW on average per trial period and in two separate years (2019 and 2020), decreased the VW in 2018, and did not cause any significant changes in its value in 2021. In other studies, the observations of the dependence of VW on the use of fungicide were also various. Some results indicate a VW increase [15,26,27], and some results show that the values of VW in control and sprayed variants did not significantly differ. At the same time, it is noted that the nature of VW is related to the climate (meteorological conditions) and the disease severity [5], and that in years with a low disease spread, VW did not increase significantly due to fungicide application [24]. Moreover, the genetic characteristics of the cultivar can affect changes in VW, depending on fungicide application [23]. A study in Sweden revealed that a single fungicide spray (at GS 45–61) resulted in an increase in VW in 12 out of 25 research years [37].

The effect of fungicide treatment on changes in CP content and ZI compared with WG content and SC is described in more detail in the literature. In Latvia, consumers prefer grain with high protein content (for bread baking at least 12%); however, none of four fungicide application variants increased it. Similar results have been obtained in several studies confirming that CP content does not depend on fungicide sprays but depends more on climatic (meteorological) conditions at the study site (including year) [5,15,21,38]. Byamakama et al. [26] found that CP content was slightly but significantly increased by fungicide application; however, the increase depended on the conditions of the study site. The comparison of different timings (GS 60 was compared to GS 39) of fungicide sprays demonstrated that later spraying did not negatively affect the CP content [24]. Some authors link the increase in CP content in the result of fungicide application to both the increase in LGA and the control of Fusarium head blight (caused by *Fusarium* spp.) [23]. In our study, only in 2018, when CP content did not reach 12% in any of the variants

(which was connected with untypically hot and dry conditions in the vegetation period), a slight (by 0.3%) but significant decrease in CP content was observed in variants F2–F4. However, the relationship between CP and SC did not change in 2018—it was negative as it is often observed. A small decrease in CP content in wheat grain as a result of fungicide application was also found in Italy, where only two variants were compared (untreated control with fungicide application at GS 55) [34], and in Sweden in separate years [37]. Landolfi et al. [34] wrote that CP decrease affected by fungicide application is probably connected with the higher grain yield. The effect of nitrogen on CP content is well-known, and in our trial, high N rates (N180–210) ensured the highest values of CP content.

The effect of fungicide application on WG content has not been widely studied. Findings in Croatia, similarly to our results, revealed that gluten content is affected by N fertilization and the meteorological conditions in the trial year but not by fungicide (tebuconazole at GS 55) application and F $\times$ N interaction [15]. ZI is an indicator characterising the CP quality. Changes in ZI were similar to changes in the values of CP and WG over the trial period and in separate years: ZI was not affected by F and F $\times$ N but was significantly affected by N top-dressing rate and meteorological conditions of the year. The effect of fungicide application on the value of ZI has not been found in previous research [5,15] either.

Starch content is not among the traditionally evaluated indicators for food wheat grain quality. SC should be evaluated in cases when wheat grain is intended for the production of ethanol [39] or feed. Similarly to CP and WG content and ZI, SC was also significantly affected by N top-dressing rate; however, the effect was contrary to that of CP, WG, and ZI, i.e., SC decreased with the increase in N rate. Moreover, the effect of the conditions in the trial year was found significant, but fungicide application mainly did not significantly affect the SC (except in 2018).

## 5. Conclusions

Based on the present findings, it can be concluded that in all studied variants of fungicide application, the average four-year grain yield increased significantly. On the other hand, more intensive spraying did not cause a greater yield increase compared with a single spraying. A strong significant year (Y) effect was noted, and only in one year characterized by normal water supply, a significant yield increase was observed. A yield decrease was observed in one hot and dry year, but fungicide-treated variants and unsprayed control variant provided equivalent grain yields in two years. Our hypothesis that a more intensive fertilization with nitrogen also requires a more intensive fungicide application was not proved, because the mathematically significant F $\times$ N interaction was small, its effect did not reveal any regularities, and the yield increase was irregular. Since a significant F $\times$ Y interaction effect was also found, it can be concluded that the choice of fungicide application should be related to the spread of diseases, which are dependent also on the year's meteorological conditions. On average, the increase in the rate of N top-dressing up to N180 kg ha$^{-1}$ increased the grain yield significantly. However, the results obtained revealed the importance of meteorological conditions. The wheat straw yield was not affected by F application and N top-dressing rate but mostly depended only on the year's conditions.

The values of TGW and VW of winter wheat on average per four-year research period were significantly affected by fungicide application and especially by the meteorological conditions of the research year, as well as by the interaction between both factors (F $\times$ Y). The N top-dressing rate and F $\times$ N interaction did not affect significantly ($p > 0.05$) both TGW and VW. Contrary was the effect of the studied factors on grain quality chemical indicators CP, WG, ZI, and SC—they were affected significantly by N top-dressing rate and the conditions of the research year. The highest N rates provided the best values of CP, WG, and ZI and the smallest values of SC. The F $\times$ N interaction did not affect significantly any of wheat grain chemical quality indicators.

**Author Contributions:** Conceptualization, A.S. (Aigars Sutka), B.B. and Z.G.; methodology, Z.G., B.B. and A.S. (Aigars Sutka); formal analyses, I.A.; investigation, Z.G., B.B., I.P.-P., L.S., J.K., G.B. and A.S. (Agrita Svarta); data curation, Z.G., B.B., A.S. (Aigars Sutka) and I.A.; writing—original draft preparation, Z.G.; writing—review and editing, Z.G., B.B., I.A. and A.S. (Aigars Sutka). All authors have read and agreed to the published version of the manuscript.

**Funding:** This research was funded by the EIP-AGRI project "The development of the decision-making support system for restriction of the diseases, affecting leaves and ears of winter wheat", project No. 18-00-A01612-000003.

**Institutional Review Board Statement:** Not applicable.

**Informed Consent Statement:** Not applicable.

**Data Availability Statement:** The data presented in this study are available on request from the corresponding author. The data will be publicly available after the project is fully completed.

**Conflicts of Interest:** The authors declare no conflict of interest. The funders had no role in the design of the study; in the collection, analyses, or interpretation of data; in the writing of the manuscript; or in the decision to publish the results.

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
