# Peer review of "Performance of Winter Wheat (Triticum aestivum) Depending on Fungicide Application and Nitrogen Top-Dressing Rate"

_agronomy, doi:10.3390/agronomy13020318_

Round 1
Reviewer 1 Report
This paper does a good job of showing the response of wheat to inputs, such as fertilizer and fungicide, can be very weather dependent. The yield limiting factors in a given year may not be N fertilizer or diseases. For example, lack of moisture and/or too extreme of temperatures may inhibit the ability of the crop to respond to or need certain inputs.
The use of “variant” when referring to the treatment factors of the studies may need to be addressed. The initial thought when variant is used is to some form of a biological organism, such as a new variant of a virus. Using “treatment” would help make it easier to read and follow.
Specific comments:
P3 L105: Shouldn’t “soil reaction” be “soil pH”? I don’t know what soil reaction 6.4-7.0 is referring to, but guessing pH.
P3 L108: When referring to plot size the dimensions are generally given (width x length). Also, the number of rows and row spacing the wheat is planting should be included.
P3 L114: What scale of wheat development is being used? It looks like Zadoks, but it should be referenced when first used in the paper to avoid confusion.
P4 L151: The company name and headquarters location need to be included so someone can duplicate the experiment.
P4 L152: Wouldn’t “applied” rather than “given” be better?
P4 L 154: What was the area harvested? Was it the middle so many rows of the plot to leave a border, or was the whole plot harvested for yield?
P4 L176: Need company name and headquarters location listed for analyzer.
P4 L178: I believe what “VW” is referring to maybe more commonly called “test weight”.
P4 L178: Some explanation of “Zeleny index” should be included for those not familiar with wheat milling/flour, especially with European terms.
P4 L179: There needs to be a reference for “LVS EN ISO 180 520:2011”. My google search was not coming up with it.
P6 L229: Correct spelling is “supply”.
P6 L 237: An example where the term “variant” is confusing, and where “treatment” would clearer.
P6 L247: Correct spelling is “green”.
P7 L267: Table 2 needs units of measure for the yield.
P7 L275: This sentence needs to be reworded. The phase “clear regularities…” adds to the confusion.
P11 L387: Where are the data for the AUDPCs? A table or a graph would be great.

Author Response
Please, see the attached file.

Reviewer 2 Report
The manuscript mainly addressed the effects of fungicide application and nitrogen top-dressing rate in winter wheat, the result will provide some key information for the cultivation of winter wheat, however, there are some problems to be revised as below.
1. In the part of Result, it will be better if the authors can change Fig 3 to one Table.
2. The effect of spraying fungicide and nitrogen top-dressing rate are not the same in four-year experiment, why? The authors should give some reasons.
3. The format of all Tables are not the same in the manuscript, the authors should change to the same format.
4. The format of some references articles are not the same, the authors should revised them.
In my opinion, the manuscript can be accepted after major revision.
Author Response
Please, see the attached file.

Reviewer 3 Report
The authors conducted a two-factor field trial with winter wheat over four years and made numerous measurements in it. The disadvantage was that in 2018 there was an extreme drought, which had an unfavorable influence on the overall picture of the results. The data were obtained and evaluated with great diligence. However, the reader (reviewer) will find numerous ambiguities and gaps in the manuscript, which I have addressed in detail in my review (see Appendix). Unfortunately, because of these deficiencies, I cannot recommend the manuscript for acceptance. Continued success in your work. BH

Author Response
Please, see the attached file.

Reviewer 4 Report
Dear appreciated Authors,
The manuscript “Performance of Winter Wheat (Triticum aestivum) Depending on Fungicide Application and Nitrogen Top-dressing Rate” was found interesting, appears to be scientifically sound and the topic gives important information related to the influence of fungicide and Nitrogen Top-dressing application on grain yield and quality of grain yield of Winter Wheat.
However, there are a few details which should be considered (see below) and some revisions have to be made and before it can reach a publishable value.
I suggest another title of manuscript (if it is appropriate time to change): "The influence of Fungicide Application and Nitrogen Top-dressing Rate on the Performance of Winter Wheat (Triticum aestivum L.)"
Line 13-15: The aim of this study was to estimate the effect of five variants of fungicide application and four levels of N (nitrogen) top-dressing rate on grain yield and quality of winter wheat.
Line 16: seasons (instead of years)
Line: 19 - 20: individual years. The yield increased significantly in one year, decreased significantly in another year, while the application of fungicides had no significant effect on the yield in remaining two seasons.
Line 21 - 25: The interaction between both examined factors was significant and the use of higher N rates not always means that spraying with fungicides have to be also more intensive. The effect of fungicide application was observed on 1000 grain weight and volume weight, while the effect of N top-dressing rate was observed on the crude protein, wet gluten and starch content and Zeleny index.
Line 26: Keywords: winter wheat; yield; grain quality; fungicides; nitrogen (it was too much keywords)
Line 30: Wheat (Triticum aestivum L.) is the one among the three main cereals which ensure food for the increasing population.
Line 52: The pressing target among farmers in Latvia is not whether to use a fungicide for winter wheat, but the real time application and its amount has been of great importance.
Line 86: GS (growth stages?)
Line 87: Therefore, the main objective of this study was to determine the effect of different intensities of fungicide application and the effect of N top-dressing rates on winter wheat grain yield and quality. Another objective was to test the hypothesis which stands that an increase in N fertilization rates requires an intensified winter wheat disease control.
Line 114: GS (growth stages?)
Line 131: Nitrogen top-dressing variants which were applied:
Line 143: The agrotechnology used in the trial was to the standard methodology, typical for this region was, while the precrop was wheat.
Line 221: Temperatures and precipitations in different growth stages caused a critical lack of moisture which was observed in the whole season of 2017/2018 and 2018/2019, as well as from the GS 49 to the harvesting maturity in 2020/2021 (HTC below 1 in Figure 2).
Line 247: during the late milk grow stage (GS 77)
Line 388: flowering
Line 389: The results showed that in 2019, when tan spot achieved the highest level, the severity of tan spot was 2 % at GS 71, while two weeks later (GS 77) it achieved 19 % in the untreated variant.
Line 395: The results revealed that the influence of meteorological conditions during the examined years showed the greater influence on the grain yield of wheat than the studied factors.
Line 458: The effect of the studied factors on both groups of grain quality indicators, both physical (TGW and VW) and chemical indicators (CP, WG, ZI, and SC) were found to be different.
Line 519: Beside in Latvia, consumers prefer grain with high protein content, at least 12 %, none of fungicide application variants increased it.
Line 538: The effect of fungicide application on WG content has not been widely studied.
Line 552: The effect of the conditions in the trial year was found significant, but fungicide application did not significantly affect the SC.
Line 555: Based on the present findings, it can be concluded that in all studied variant of fungicide application the average four-year grain yield increased significantly. On the other side more intensive spraying did not cause a greater yield increase compared with a single spraying.
Line 564: F x Y?
Line 567: However, the results obtained here, revealed the importance of climatic conditions. The wheat straw yield was not affected by F application and by N top-dressing rate, while mostly depended on the year’s conditions.
Line 567: However, the results obtained here, revealed the importance of climatic conditions. The wheat straw yield was not affected by F application and by N top-dressing rate, while mostly depended on the year’s conditions.
Line 569: The values of TGW and VW of winter wheat on average per four-year research period were significantly affected by fungicide application and specially by the weather conditions of the research year, as well as from the interaction between both factors (F × Y).
Kind regards,
NL

Author Response
Please, see the attached file.

Round 2
Reviewer 3 Report
Z 19-20: This statement that yields vary from year to year is very simplistic and needs no investigation. Should be deleted from the summary.
Z 23-26: What kind of effect is meant here (negative, positive, significant)? Suggestion: "A clear effect of ...was observed ..." "... while N top-dressing led to significant changes of ..."
Z 41: "demanding crop" (too general), demanding for what?
Z 151: Is there data on soil mineral N (NO3, NH4) content? This is important in field trials for N fertilization.
Z 155: Which kind of growth regulator was applied in the trial (trade name, chemical compound)?
Z 182: Here VW and TGW are named with abbreviations, but the other parameters are not named with abbreviations: crude protein (CP), wet gluten (WG), zeley index (ZI), dry matter (DM). Authors should refer to all parameters with abbreviations and use this consistently throughout the text.
Z 235: I understand that the authors want to present all the data they collected in the experiment. But it is better to present only the results that fit the objective and hypothesis. N fertilization and fungicides had no effect on wheat development. Therefore, this figure is also not relevant.
Z 241: The HTC values do not belong in the results section. They describe the weather conditions during wheat development and are therefore part of the experimental conditions (method) and not part of the results.
Z 245-Z254: This text does not belong in the results section.
I would replace the term "moisture" with "water" throughout the text.
Z 264: The authors should write/explain which disease is meant by "Tan spot". Is it "Pyrenophora tritici-repentis"?
Z 402: In the title of the table, the authors should indicate that the quality parameters were measured by NIRS. This is important because it is possible to determine these parameters by other methods to (e.g. protein by "Kjeldahl method").
Z 582- 583: The results are only described and compared with results from other countries. However, the authors do not give any reasons (possible explanations) for the significant decrease in CP. Did the relationship between CP and starch content in the seeds change (negative correlation between CP and starch)?
Z 616-617: This conclusion is correct. But how to implement it? Here the authors should explain how the fungicide application should adapt to the annual weather.
Z 623-631: This paragraph only summarizes results related to TGW, VW and quality parameters. No conclusion is drawn here. It is not useful to summarize only the results in the Conclusions section, which is redundant.
Other comments:
Titles of figures and tables: A basic rule in scientific writing is to mention in the title of the figures/tables the experiment, location, and year to which the results refer. Although the method describes where and how the experiments were performed, authors should include these details.
For example: Table 2. Development of wheat leaf diseases depending on fungicide treatment and year, field experiment with winter wheat 2018-2021, research farm Peterlauki, Latvia University of Life Science and Technology (data in AUDPC units).
Figures 3 to 5: The information on the scatter of the individual values around the mean value (standard error or standard deviation) is missing here. In international publications it is common to indicate these scatter values.
In the bibliography, the page numbers are sometimes designated with "pp", but sometimes "pp" is omitted. This should be done consistently.
Author Response
Dear reviewer!
Thank you for your review and valuable comments. We have revised our manuscript according the new advice given in the second review round. You will find a point-by-point response to every comment in the attached file. All the corrections are visible in the manuscript by Tack Changes tool.
Sincerely,
Zinta Gaile
